# Production of Terpene Trilactones from Cell and Organ Cultures of *Ginkgo biloba*

**DOI:** 10.3390/plants13182575

**Published:** 2024-09-13

**Authors:** Hosakatte Niranjana Murthy, Guggalada Govardhana Yadav, Kee Yoeup Paek, So-Young Park

**Affiliations:** 1Department of Botany, Karnatak University, Dharwad 580003, India; 2Department of Horticultural Science, Chungbuk National University, Cheongju 28644, Republic of Korea; 3Department of Biotechnology, KLE Technological University, Hubballi 580031, India

**Keywords:** cell cultures, *Ginkgo biloba*, elicitation, immobilization, secondary metabolites, terpenoids

## Abstract

*Ginkgo biloba* is an ancient plant that has survived up until the present day. *Gingko biloba* is a rich source of valuable secondary metabolites, particularly terpene trilactones (TTLs) such as ginkgolides and bilobalides, which are obtained from the leaves and seeds of the plant. TTLs have pharmacological properties, including anticancer, anti-dementia, antidepressant, antidiabetic, anti-inflammatory, anti-hypertensive, antiplatelet, immunomodulatory, and neuroprotective effects. However, ginkgo is a very-slow-growing tree that takes approximately 30 years to reach maturity. In addition, the accumulation of TTLs in these plants is affected by age, sex, and seasonal and geographical variations. Therefore, plant cell cultures have been established in ginkgo to produce TTLs. Extensive investigations have been conducted to optimize the culture media, growth regulators, nutrients, immobilization, elicitation, and precursor-feeding strategies for the production of TTLs in vitro. In addition, metabolic engineering and synthetic biology methods have been used for the heterologous production of TTLs. In this review, we present the research strategies applied to cell cultures for the production of TTLs.

## 1. Introduction

The gymnosperm plant *Ginkgo biloba* L. (Ginkgoaceae) (Figure 1), also known as the “maidenhair tree” or “ginkgo”, has endured for several million years. According to fossil evidence, the *Ginkgo* genus was extant approximately 170 million years ago [1]. Ginkgo is frequently referred to as a “living fossil”, since its morphology seems to have changed relatively little over more than 100 million years [1]. Ginkgo trees can survive for more than a millennium. *Ginkgo biloba* is dioecious, i.e., it has separate sexes (Figure 1). Plants take approximately 30 years to reach maturity; the maximum height and diameter of their main stems are 35 and 10 m, respectively. The trees can withstand pollution and are not significantly affected by insects, bacteria, or fungi. *Ginkgo biloba* is commonly found in China, Korea, and Japan and is cultivated in various parts of the world, especially in botanical gardens [2]. The leaves are fan-shaped, with veins radiating into the leaf blade and sometimes bifurcating (splitting) (Figure 1). The leaves resemble the pinnae of the maidenhair fern (*Adiantum* species); therefore, they are named maidenhair trees [2].

Ginkgo has been utilized in traditional Chinese medicine since the eleventh century. Ginkgo seeds and leaves are used in traditional medicine to treat various diseases, including renal problems, asthma, bronchitis, and dementia. In the herbal market, a *Ginkgo biloba* leaf preparation known as “EGb761” is marketed under the names Tanakan, Rokan, and Tebonin Forte for the treatment of memory-related conditions [3].

The major phytoconstituents isolated from *Ginkgo biloba* include terpenoids, flavonoids, carboxylic acids, lignins, proanthocyanins, polyprenols, polysaccharides, alkylphenols, and alkylphenolic acids [3,4]. The most significant phytoconstituents for the prevention and treatment of cardiovascular and cerebrovascular illnesses are terpene trilactones (TTLs), namely bilobalides (sesquiterpenes) and ginkgolides (diterpenes) [4]. Ten diterpenoid lactones, ginkgolides A, B, C, J, K, L, M, N, P, and Q, have been previously identified (Table 1 and Figure 2). Bilobalide is a member of the sesquiterpene lactone group, and its isomer has two lactone ring groups (Figure 2). Ginkgolides A, B, C, J, and M (about 2.8–3.4%) and bilobalides (2.6–3.2%) constitute the majority of the terpenoid fraction of the extract. The TTLs of ginkgo exhibit a variety of biological activities; for instance, ginkgolide A has been shown to have immunostimulatory and anti-inflammatory properties [5]. According to previous reports, ginkgolide B affects the central nervous system, platelet activation, and antioxidant defense and has anti-inflammatory and anti-apoptotic effects [6,7]. DeFeudies et al. [8] reported that ginkgolide B inhibits the proliferation of human breast cancer cell lines. Ginkgolide C has been shown in earlier studies to have anticancer effects and to lower hyperlipidemia [9]. Bolshakov et al. [10] reported that ginkgolide M stimulates the central nervous system, whereas Vitolo et al. [11] found that ginkgolide J has anti-dementia activity. Ginkgolide K has antioxidant, immunomodulatory, and neuroprotective properties [12,13]. Zhao et al. [14] showed that ginkgolide N inhibits cholinesterase activity. In contrast, bilobalide exhibits antioxidant, cardioprotective, neuroprotective, anti-inflammatory, and anti-ischemic properties [15]. TTLs of ginkgo are in high demand in the pharmaceutical and nutraceutical industries because of the various biological actions that have been previously discussed. However, *Ginkgo biloba* is a slow-growing tree that requires labor-intensive care. Moreover, seasonal changes, climate, and geographic location affect TTL production in vivo [16]. Plant secondary metabolites can be produced in large quantities using biotechnological techniques such as organ and cell cultures. Numerous studies using cell and organ cultures have been conducted to produce TTLs. Therefore, this review aims to provide information on the in vitro culture of cells and organs used in the production of ginkgo terpenoids. The cloning of advantageous terpenoid biosynthetic genes into bacteria and yeast, which can produce more terpenoids through metabolic engineering, is also discussed.

**Table 1 plants-13-02575-t001:** Terpene trilactones of *Ginkgo biloba* and their biological activities.

Name of the Compound	Biological Activities	References
Diterpenes		
Ginkgolide A	Anti-inflammatory and immunostimulant	[5]
Ginkgolide B	Stimulation of the central nervous system	[7]
Ginkgolide C	Antilipidemic and anticancer	[9]
Ginkgolide M	Central nervous system activity	[10]
Ginkgolide J	Anti-dementia	[11]
Ginkgolide P	No data	[4]
Ginkgolide K	Antioxidant, immunomodulatory, and neuroprotective	[13,17]
Ginkgolide Q	No data	[18]
Ginkgolide L	No data	[19]
Ginkgolide N	Inhibition of cholinesterase activity	[14]
Sesquiterpenes		
Bilobalide	Anti-inflammatory, anti-ischemic, antioxidant, cardioprotective, and neuroprotective	[15]
Bilobalide isomer	No data	[20]

In higher plants, terpenoids, including TTLs, are synthesized via distinct mevalonate (MVA) and 2-C-methyl-D-erythritol 4-phosphate (MEP) pathways [17]. The MVA pathway occurs in the cytosol and is initiated by acetyl-CoA, and the enzyme acetoacetyl-Co-A thiolase (AACT) condenses two molecules of acetyl-CoA to give acetoacetyl-CoA. In the subsequent step, HMG-CoA synthase (HMGS) condenses acetyl-CoA and acetoacetyl-CoA to produce 3-hydroxy-3-methylgultaryl-CoA (HMG-CoA). After that, HMG-CoA is reduced by HMG-CoA reductase (HMGR) to produce mevalonic acid. The following stages sees the production of dimethylallyl diphosphate (DMAPP) and IPP. Mevalanate kinase (MVK) and mevalanate-5-phospahte (PMK) phosphorylate mevalonic acid twice at the 5′-OH site before mevalanate-5-diphopshate decarboxylase (MVD) decarboxylates it to isopentenyl diphosphate (IPP) and its isomer DMAPP (Figure 3) [21]. In the subsequent step, farnesyl phyrophosphate (FPP) is synthesized by the activity of the enzyme farnesyl pyrophosphate synthase (FPS). It was proposed that the sesquiterpene bilobalide is thought to be derived from farnesyl pyrophosphate [22].

The MEP pathway occurs in plastids, which leads to the formation of isopentenyl diphosphate (IPP) and dimethylallyl diphosphate (DMAPP) [23]. In this pathway, 1-deopy-D-xylolose 5-phospahte synthase (DXS) catalyzes the condensation of glyceraldehyde-3-phosphate and pyruvate to form 1-deopy-D-xylolose 5-phospahte (DXP), which is then reduced by DXP reductoisomerase (DXR) to form MEP. 2-C-methyl-D-erythritol 4-phosphate cytidyltransferase (MECT), another enzyme, catalyzes the formation 4-(cytidine 5′-diphospho)-2-C-methyl-D-erythritoal from MEP. The following two steps are then catalyzed by 4-(cytidine 5′-diphospho)-2-C-methyl-D-erythritol kinase (CMEK) and 2-C-metyl-D-erythritol 2,4-cyclodiphosphate synthase (MECS), respectively. After that, 1-hydroxy-2-methyl-2(E)-bytenyl-4-diphosphate synthase (HDS) catalyzes 2-C-methyl-D-erythritol 2,4-cyclodiphosphate, resulting in the formation of 1-hydoxy-2-methyl-2-(E)-butenyl 4-diphenyl 2-diphenylphosphate (HMBPP). The MEP pathway culminates in the production of IPP and its isomer DMAPP by the reduction of HMBPP by HMBPP reductase (HDR). Geranylgeranyl diphosphate (GGDP) is biosynthesized from IPP and DMAPP by the catalysis of geranylgeranyl diphosphate synthase (GGPPS) [23]. In recent years, several genes of the MEP pathway, such as the HMGR and HMGS genes, have been recognized and isolated in *G. biloba* [24,25,26].

IPP and DMAPP are condensed by GGPPS to synthesize 20-carbon genranylgeranyl diphosphate (GGPP), which serves as a liner skeleton for TTL biosynthesis. GGPP is cyclized by the enzyme lvopimaradiene synthase (LPS) to form levopimaradiene (LP) and subsequently translocated from plastids to the cytoplasm [26]. The cyclization of GGPP by terpenoid synthase followed by the oxidation process by cytochrome P450 (CYP450) results in the synthesis of ginkgolide (Figure 3) [27].

## 2. Production of Terpene Trilactones in Plant Tissue Cultures

Plant cell and organ cultures require the use of a variety of strategies such as the selection of cell lines, the right nutrient medium, the strength of the nutrients, growth regulators, medium constituents such as the type and source of nitrogen and the concentration of sucrose, and physical factors such as light and temperature for biomass and secondary metabolite production [28,29,30,31,32,33,34], all of which have been well established with cell cultures of *Ginkgo biloba*. Different strategies for producing TTLs from ginkgo cell cultures are shown in Table 2 and discussed in the following various sections.

### 2.1. Selection of Cell Lines

Park et al. [35] assessed the concentrations of TTLs, including bilobalide, ginkgolide A (GK-A), and ginkgolide B (GK-B), in several organs and in individuals descended from *Ginkgo biloba* plants that were male and female. According to their findings, large amounts of GK-B and bilobalide were found in the leaves of female trees, whereas GK-A was found in greater amounts in stems and stem bark, with traces of both GK-B and bilobalide. GK-A was found to be generally consistent and ranged between 200–300 µg/g dry weight (DW). However, there was significant fluctuation in GK-B and bilobalide, with values ranging from 800–1000 µg/g and less than 100 µg/g, respectively. GK-A was the predominant type in all the plant parts of male ginkgo trees. Compared to the GK-B and bilobalide contents in the female trees, the GK-A content of the leaves was lower. Therefore, it is essential to select the correct organ and superior accession for callus regeneration.

Using calli taken from the petioles of both male and female trees, Park et al. [35] established cell cultures in Murashige and Skoog’s (MS) [36] medium supplemented with 20 µM naphthalene acetic acid (NAA) and 3% sucrose for four weeks. They observed that the predominant component in the suspension cultures derived from both male and female trees was GK-A. Additionally, TTLs accumulated in cell cultures in the following order: GK-A > bilobalide > GK-B, based on their respective concentrations.

Balz et al. [37] conducted a study wherein they established cell cultures with 80 cell strains and transformed root cultures using *Agrobacterium rhizogenes*. The results showed that the levels in the cell cultures were 1 µg/g DW and 4 mg/g DW in the transformed hairy root cultures. Laurain et al. [38] established in vitro cultures utilizing callus cultures from zygotic embryo cultures, prothalli, microspore-derived cell suspensions, and transformed cell cultures (using *Agrobacterium rhizogenes*). The transformed cell cultures contained ginkgolides and bilobalide, but the cell cultures derived from the prothallus contained more ginkgolides. However, ginkgolides and bilobalides were absent from the callus cultures and microspore-derived cells isolated from zygotic embryos. These data demonstrate the process of selecting the best cell line to produce TTLs from cell suspension cultures.

### 2.2. Optimization of Nutrient Medium

Jeon et al. [39] established callus and cell suspension cultures in ginkgo to produce TTLs. They studied the effects of various media on the growth of calli and cell suspensions. Among all the tested media, MS [36] and Schenk and Hildebrandt’s (SH) media [40] were found to be superior for the accumulation of fresh and dry cell biomass.

Jeon et al. [41] evaluated the effects of MS and SH media supplemented with 1.0 mg/L NAA, 0.1 mg/L kinetin, and 30 g/L sucrose. They found that the cells cultivated in the MS medium showed the maximum induction of the GK-B content, whereas the cells in the SH medium showed increased cell proliferation. As a result, they decided to use the MS medium in their upcoming research.

In another investigation, Park et al. [42] examined the impact on biomass and ginkgolide accumulation in a ginkgo cell suspension culture, and MS medium outperformed the other culture media in terms of both cell proliferation and ginkgolide production. These results indicated the impact of the nutrient content on the generation of biomass and secondary metabolites.

### 2.3. Effect of Plant Growth Regulators

Growth regulators are usually supplied exogenously to cell and organ cultures to promote biomass growth, proliferation, and secondary metabolite production. In general, the type and quantity of plant growth regulators play critical roles in the development of cells and organs and the accumulation of metabolites.

Jeon et al. [39] investigated the effect of NAA in the range of 1–8 mg/L to confirm its influence on biomass and metabolite accumulation. The concentration of NAA in the medium was found to encourage cell proliferation in the range of 1.0–4.0 mg/L; however, cell growth declined at concentrations higher than 4 mg/L. They employed MS medium supplemented with the aforementioned plant growth regulator combination for the production of TTLs since the cell growth was optimal in the medium supplemented with 0.1 mg/L NAA and 0.1 mg/L kinetin [41]. Carrier et al. [43] and Huh and Staba [44] used a combination of NAA and kinetin to establish ginkgo cell cultures.

Park et al. [42] investigated the effects of 5, 10, 20, and 40 µM of NAA or 2,4-D on biomass accumulation in ginkgo. According to their findings, 20 µM of 2,4-D and NAA was the effective concentration needed to induce and develop calli. Furthermore, the callus that grew on MS medium supplemented with 2,4-D was brownish, whereas the callus that regenerated on medium supplemented with NAA was white or yellow. They also observed poor quality maintenance in calli regenerated on a 2,4-D-containing medium. After realizing that the callus regenerated on a medium containing 20 µM NAA was beneficial for growth and biomass accumulation, Park et al. [42] established cell suspension cultures in small-scale and bioreactor cultures using this medium.

### 2.4. Impact of Nitrogen and Phosphate Levels

Ginkgo cell suspension cultures were established as described by Carrier et al. [43] using MS medium supplemented with 1 mg/L NAA, 0.1 mg/L kinetin, and 30 g/L sucrose in flasks and bioreactors (2 and 6 L capacities). After 27 days of culture, the quantities of extracellular phosphate and nitrate in the spent media were evaluated. The cultivated cells did not fully utilize the 40 mM nitrate in the MS medium; 10 mM extracellular nitrate remained unconsumed at the end of the culture (after 27 days). Ammonium and phosphate were consumed by day 26 in the bioreactor cultures at an average rate of 0.63 mM/day and 0.002 mM/day, respectively. In a different study, Jeon et al. [41] established cell suspension cultures of ginkgo using MS medium with modified molar ratios of ammonium and nitrate ions (5/1, 3/1, 1/1, 1/2, 1/3, 1/5, and 0/6) and KH_2_PO_4_ (0, 0.25, 0.75, 1.25, 1.75, and 2.25 mM), and the optimum dry biomass and ginkgolide contents were obtained with 1/3 ammonium and nitrate ion concentrations and 1.25 mM KH_2_PO_4_ concentrations. These results demonstrate that higher concentrations of nitrate and ammonium are useful for the production of ginkgo biomass and ginkgolides.

### 2.5. Influence of Sucrose Concentration

Jeon et al. [41] investigated the effect of sucrose concentration on the culture medium at concentrations of 20, 30, 40, and 60 g/L. They found that the maximal cell growth was achieved in the media supplemented with sucrose concentrations ranging from 30–40 g/L; they reported a decline in cell growth at a sucrose concentration of 60 g/L. Park et al. [42] studied the influence of 1, 3, 5, and 7% sucrose supplemented in MS medium on the growth and accumulation of ginkgolides in cell suspension cultures of ginkgo. Their study revealed that the supplementation with 3% sucrose in the medium favored biomass accumulation, whereas higher sucrose levels (5 and 7%) improved ginkgolide production.

### 2.6. Inoculum Density, Light/Dark, and Temperature Effects

Park et al. [35,42] used an MS medium supplemented with 20 µM NAA and 3% sucrose to study the effects of the inoculum density, light/dark cycle, and temperature in ginkgo cell cultures. In terms of sediment volume, they examined the effects of 10, 20, 30, 40, and 50% inoculum. They discovered that a 20% cell inoculum was the most successful and reached the stationary phase after three weeks [42]. Park et al. [35] studied the effects of continuous dark and a 16 h light and 8 h dark photoperiod on ginkgo cell cultures grown from male and female plants for 4 weeks. They observed differences in the production of GK-A, GK-B, and bilobalide. Compared to the original sexuality of the cells (derived from either female or male plants), the production of GK-A was more abundant in the female cells under light conditions, and the opposite was true in the male cells. In contrast, the male cells barely produced GK-B under both light and dark conditions. Furthermore, Park et al. [35] observed that the production of GK-B in the female cells was greatly elevated in response to light, increasing by 1 mg/g DW. These findings show that the modification of the metabolism by light irradiation is also expected depending on individual variation. Additionally, Park et al. [35] observed that bilobalide synthesis in both male and female cells was enhanced by darkness as opposed to light, with cultures grown in darkness exhibiting up to 3- and 4.7-fold increases in bilobalide production, respectively, under light conditions.

Park et al. [42] investigated the effect of temperature (20, 24, 28, 32, and 36 °C) on biomass and secondary metabolite synthesis in ginkgo cell suspension cultures. They discovered that the best temperature for developing cells was determined to be 25 °C; otherwise, rapid declines in cell growth were observed either below or above the optimal temperature. Furthermore, they observed that the best conditions for the production of GK-A and bilobalide were observed in cell cultures grown at high temperatures (36 °C). These findings imply that cell cultures can be cultivated at lower temperatures until a substantial amount of biomass accumulation is obtained, at which point they can be incubated at 36 °C to promote the production of secondary metabolites.

**Table 2 plants-13-02575-t002:** Successful examples of terpene trilactone production from in vitro cell and organ cultures.

Type of Culture	Medium Composition	Strategy Followed	Response	Total Ginkgolide Content	References
Cell cultures	Murashige and Skoog (MS) medium with 1 mg/L naphthaleneaceticacid (NAA), 0.1 mg/L kinetin and 30 g/L sucrose	Immobilized cells were cultured in 2 and 6 L bioreactors and shake flasks	The biomass yield was 14.24 and 14.82 g/DW/L for the 2 and 6 L immobilized cultures	7, 17, 19, and 7 ng/g DW amounts of ginkgolide A were obtained in a shake flask (500 mL), shake flask (1200 mL), 2 L, and 6 L immobilized bioreactor cultures, respectively	[43]
		The extracellular levels of phosphate, nitrate, ammonium, and carbohydrates were studied			
Callus and cell cultures	Modified MS medium with 30 g/L sucrose	The effect of different media was tested on induction callus, callus growth, and cell suspensions	The MS media with 1.0 mg/L and 0.1 mg/L NAA was excellent for callus induction	Not reported	[39,41]
		The effect of NAA (0.5, 1, 2, 4, and 8 mg/L) was studied	The growth of the cells was good on MS medium with 1.0 mg/L NAA, 30 g/L sucrose, 1.75 mM phosphate, and a 1:5 molar ratio of NH_4_^+^ to NO_3_^−^.		
		The effect of the concentration of sucrose (20, 30, 40, and 60 g/L) was studied			
		The effect of the molar ratio of ammonium and nitrate ions (5/1, 3/1, 1/1, 1/2, 1/3, 1/5, 0/6 was studied			
		The effect of the molar concentration of KH_2_PO_4_ (0, 0.25, 0.75, 1.25, 1.75, 2.25 mM) was studied			
Cell culture	MS medium with 2 mg/L NAA and 0.2 mg/L kinetin or 2 mg/L benzyladenine (BA)	Cell cultures transformed by *Agrobacteriumrhizogenes*	Transformed cell cultures possessed ginkgolides and bilobalide	Transformed cells contained 147, 83, 137, 87, and 200 µg/g DW of ginkgolide A, B, C, J, and bilobalide.	[38]
		Cell suspension derived from the prothallus	Cell cultures of prothallus-derived cells contained higher ginkgolide levels	Prothallus-derived cells possessed 270, 160, 213, 70, and 160 µg/g DW of ginkgolide A, B, C, J, and bilobalide.	
		Callus from a zygotic embryo	No ginkgolides or bilobalides with microspore-derived cells and calli from immature zygotic embryos		
		Cell suspension derived from microspores			
Embryo germination	MS medium with 1.0 mg/L thiamine chloride, 100 mg/L myoinositol, and 30 g/L sucrose	Comparison of ginkgolides in seeds, embryos, albumen, seedlings, and plantlets germinated in vitro	Plantlets germinated in vitro possessed highest ginkgolide levels	1250 and 844 µg/g DW ginkgolide A and ginkgolide B	[45]
Cell and hairy root cultures	MS medium with 30 g/L sucrose or 20 g/L glucose‘	Cell cultures were established by using different strains	Transformed roots possessed terpenes at the same concentration as whole plant leaves; however, they had a very slow growth rate	From 0.2 to 6.0 mg/g DW	[37]
		Transformation of ginkgo tissue using *Agrobacterium rhizogenes*			
Cell culture	MS medium with 20 µM NAA and 100 mg/L myoinositol	Comparison of cell lines derived from male and female plants with that of in vivo leaves, stem bark, and stems	Variability in ginkgolide and bilobalide contents of ginkgo lines, sexuality, and the plant parts	619, 976, and 198 g/g DW of GK-A, GK-B, and bilobalide were obtained in the light-exposed female line as compared to 184, 32, and 587 g/g DW of GK-A and GK-B in the continuous-dark-grown female line	[35]
		The effect of light irradiation (16 h photoperiod) vs. continuous dark incubation of cultures on ginkgolide and bilobalide accumulation	Ginkgolide (GK)-A was the major constituent in suspension cultures of both male and female tree lines. Furthermore, accumulation was in the following order: GK-A > bilobalide > GK-B. More bilobalide was recorded in the male line than in the female line		
			Light provoked a higher accumulation of GK-A and GK-B; however, the concentration of bilobalide decreased		
Cell culture	MS medium with 20 µM NAA and 100 mg/L myoinositol	The effect of MS medium supplemented with 20 µM NAA and 100 mg/L myoinositol was tested	MS medium with 20 µM NAA was good for cell growth and ginkgolide production	6.5, 0.5, and 3.5 mg/L ginkgolide A, ginkgolide B, and bilobalide	[42]
		The effect of 1, 3, 5, and 7% sucrose levels was tested	Of the various levels of sucrose tested, 3% sucrose enhanced cell growth; however, the media supplemented with 5 and 7% sucrose increased bilobalide and ginkgolide A production		
		The effect temperatures of 20, 24, 28, 32, and 36 °C on cell growth and ginkgolides was tested	Among the various temperatures tested, 25 °C was good for biomass accumulation, and 36 °C increased bilobalide and ginkgolide A production		
		Balloon-type bubble bioreactors	Among the various temperatures tested, 25 °C was good for biomass accumulation, and 36 °C increased bilobalide and ginkgolide A production		
			In a 5 L balloon-type bubble bioreactor containing 2.5 L medium, good metabolite production was realized		

## 3. Application of Immobilization, Elicitation, and Precursor Feeding Strategies

Plant cells can be immobilized in appropriate matrices to overcome the problems of cell aggregation and low shear resistance. Cell immobilization has numerous advantages, including increased product accumulation, reduced shear stress, simpler downstream processing, and extended cell viability. A regular technique in plant cell culture involves the application of elicitors, such as bacterial or fungal cell wall materials (polysaccharides or glycoproteins), higher concentrations of salt or heavy metals, nanomaterials, or signaling molecules such as salicylic acid or methyl jasmonate. These methods have been used to enhance secondary metabolite production in cell suspension cultures [32,46]. Achieving optimal results requires careful consideration of the elicitor concentration, exposure length, and culture stage at the time of elicitor application. Feeding precursors, which are substances involved in the biosynthetic pathway of the target compounds, are an additional strategy used to achieve increased secondary product accumulation. Table 3 shows how the abovementioned strategies have been used to produce TTLs in ginkgo cell suspension cultures.

Kang et al. [47] used ginkgo cell suspension cultures to examine the effects of KCl treatment (using salt as an elicitor) at concentrations of 50, 200, and 800 mM. The cultivated cells produced more GK-B, GK-A, and bilobalide due to salt stress, and these metabolites were discharged into the culture medium. Although the treatment of the cell cultures with 800 mM KCl decreased cell growth and biomass formation, it produced higher levels of TTLs. Kang et al. [47] exposed cultures to salt stress for 12 days after initiation. GK-A and GK-B production increased by 1.9 and 4.0 times, respectively, compared to the control, following the administration of 800 mM for 48 h.

In another study, Sukito and Tachibana [48] employed the immobilization of cells and used salicylic acid and methyl jasmonate as elicitors to produce GK-A, GK-B, GK-C, and bilobalide in ginkgo cell suspension cultures. Jute fibers were used to immobilize the ginkgo cells, and a combination of methyl jasmonate (0.1 mM) and salicylic acid (0.1 mM) was subsequently applied. The resulting treatment consisted of 78, 79, 71, and 7.5 mg/g DW of bilobalide, GK-A, GK-B, and GK-C, respectively. These concentrations were 1.78, 1.95, 2.05, and 2.95 times higher than the control treatments.

Lin et al. [49] investigated the immobilization of ginkgo cells using 1 cm^3^ cubes of a sponge (500 µM in diameter), cotton fiber (500 µM in diameter), and a loofah (2–3 mm in diameter), and subsequently, the immobilized cells (immobilized cells were cultured for 14 days) were elicited using chitosan (50, 100, 200, 300, 400, and 500 mg/L), yeast extract (50, 100, 200, 300, 400, and 500 mg/L), methyl jasmonate (0.01, 0.05, 0.1, 0.3, and 0.5 mM), or salicylic acid (0.01, 0.05, 0.1, 0.3, and 0.5 mM) for 2 days. The initial experimental results showed that MJ and SA were good elicitors and that cotton fiber was good for immobilization, which led to a higher accumulation of GK-B in the immobilized cells (intracellular metabolite) and the culture medium (extracellular metabolite). They demonstrated that the effectiveness of each elicitor in stimulating GK-B was in the following order: MJ > SA > YE > CH, and the effectiveness of the immobilization material was in the following order: cotton fiber > loofah > sponge. In their subsequent experiments, Lin et al. [49] tested the effect of 0.3 mM SA or 0.5 mM MJ and ginkgo cells immobilized in cotton fiber and a loofah with the treatment of elicitor while replacing the fresh medium without elicitor cycles. Through these treatments, they realized the accumulation of 114 mg/L and 1.2 mg/L of GK-B with a cycle of 4–14-day recovery. Based on their findings, they concluded that 0.5 mM MJ with cotton fiber (as an immobilization material) was the optimal combination for efficient GK-B production.

To increase ginkgolide synthesis, Zhu et al. [50] investigated the effect of levopimaradiene (LP) on cambial meristematic cells (CMCs) and ginkgo-differentiated cells (DDCs). At 12, 24, 36, 48, and 60 h, they applied 20, 40, 60, 80, 100, and 120 mg/L of LP to suspension cultures that were 13 days old. In ginkgo DDCs (with 100 mg/L LP for 48 *h*), the production of GK-A and GK-B was 1.61 and 1.32 times higher than that in the control group, respectively. Conversely, after 60 h of LP treatment, the synthesis of GK-C and bilobalide increased to 234 and 161 µg/L, respectively. When LP was applied to CMCs, the production of GK-A, GK-B, GK-C, and bilobalide was 2.03, 1.43, 1.22, and 1.19 times greater than that in the control groups, respectively.

To examine the effects of precursor feeding on ginkgo cell cultures, Kang et al. [51] supplied 12-day-old cell cultures after 5 days of treatment with various intermediates of the MVA and MEP pathways, including 0.01 mM geranyl pyrophosphate (GPP), geranylgeranyl pyrophosphate (GGPP), isopentenyl pyrophosphate (IPP), dimethylallyl pyrophosphate (DMAPP), farnesyl pyrophosphate (FPP), acetyl-CoA, 3-hydroxy-3-methylglutaryl-CoA (HMG-CoA), glyceraldehyde-3 phosphate (GA-3P), mevalonate (MVA), and sodium pyruvate (SP). The cultures treated with the different precursors had higher levels of GK-A, GK-B, and bilobalide (both intracellularly and extracellularly); however, the administration of each precursor resulted in diverse effects. Overall, Kang et al. [51] found that precursors upstream of the metabolism affected GK-A accumulation in the cell cultures, whereas downstream precursors in the terpenoid biosynthetic route affected GK-B accumulation.

**Table 3 plants-13-02575-t003:** Application of immobilization, elicitation, and precursor feeding strategies for the production of terpene trilactones from in vitro cell and organ cultures.

Type of Culture	Medium Composition	Strategy Followed	Response	Total Ginkgolide Content	References
Cell culture/ elicitation	MS medium with 3% sucrose and 3.5 mg/L NAA	50, 200, and 800 mM KCl were used as elicitors and added to 14-day-old cell suspension cultures and treated for 12, 24, 48, and 72 h	800 mM KCl severed inhibited the cell growth. However, it was responsible for 1.9- and 4.0-times higher accumulation of ginkgolide A and ginkgolide B	Elicitor was responsible for 1.9 (15 mg/g DW) and 4.0 times (8 mg/g DW) higher accumulation of ginkgolide A and ginkgolide B	[47]
Immobilized cell culture/elicitation	MS medium with 3% sucrose, 2 mg/L NAA, and 0.1 mg/L kinetin	Immobilization of cells using jute fibers	Immobilization cells lead 1.4 times higher accumulation of biomass	78, 79, 71, and 7.5 mg/g DW bilobalide, ginkgolide A, ginkgolide B, and ginkgolide C, respectively	[48]
		Eliciting with methyl jasmonate (MJ) and salicylic acid (SA) alone or in combination (0.01, 0.1 mM)	The combined elicitation (0.1 mM MJ + 0.1 SA) was responsible for 1.78-, 1.95-, 2.05-, and 2.95-fold more bilobalide, and ginkgolides A, B, and C respectively, compared to control		
Cell culture/effect of levopimaradiene (diterpene resin acid)	MS medium with 3% sucrose and 50 mg/L ascorbic acid. Two types of cell lines were used viz. dedifferentiated cells (DDCs) and cambial meristematic cells (CMCs). For the DDCs culture, 2 mg/L NAA and 2 mg/L indole butyric acid were added, while for the CMCs culture, 2 mg/L NAA and 2 mg/L 2,4-dichlorophenoxyacetic acid (2,4-D) were added	Cell cultures were treated with levopimaradiene (LP) at 20, 40, 60, 80, 100, and 120 mg/L in 13-day-old cell cultures	The productions of ginkgolide A (GA) and ginkgolide B (GB) were 1.61- and 1.32-fold larger than that of the control groups when *G. biloba* DDCs were treated with LP, and the productions of ginkgolide C (GC) and BB reached 234 and 161 μg/L after being treated with LP for 60 h.	234 and 161 μg/L ginkgolide C and bilobalide with 60 h LP treatment in DDCs	[49]
		The transcript levels of *DXS*, *MECT*, *HDS*, *HDR*, *GGPP,* and *LPS* (*GAPDH* was used as a housekeeping gene) were monitored by qRT-PCR	The production of GA, GB, GC, and BB was 2.03-, 1.43-, 1.22-, and 1.19-fold larger than that of the control groups in LP-treated CMCs		
Immobilized cell culture/ elicitation	Half-strength MS medium with 2 mg/L NAA, 0.5 mg/L kinetin, 0.1 g/L casein hydrolysate, and 0.8 g/L polyvinyl pyrrolidone	Cell suspension (2 mg) in 20 mL 1/2 MS medium with chitosan (CH, 50, 100, 200, 300, 400, and 500 mg/L) or yeast extract (YE, 50, 100, 200, 300, 400, and 500 mg/L) or methyl jasmonate (MJ, 0.01, 0.05,0.1, 0.3, and 0.5 mM) or salicylic acid (0.01, 0.05,0.1, 0.3 and 0.5 mM) was tested as elicitors.	The ginkgolide B (GB) intercellular and extracellular yields were 108.9/112.4 mg/L with 0.5 mM MJ treatment; whereas GB intercellular and extracellular yields were 7.4.4/82.1 mg/L with 0.3 mM SA treatment. Thus, increment of GB yield 371%/404% and 222%/268% with MJ and SA treatments	114 mg/L GB could be produced with cotton immobilization of cells and 0.5 mM MJ treatment	[50]
		Based on the above experiment, 0.5 and 0.3 mM MJ and SA were found to be the optimum concentrations for GB production; therefore, the subsequent cell cultures were established as above, and the cells were immobilized with 1 cm^3^ cubes of sponge, cotton fiber (pore size 500 µm in diameter), and a loofah (2–3 mm in diameter) for 14 days	The immobilization of cells with three materials showed that cotton and loofah were good in terms of cell quality, viability, and GB production		
Precursor feeding					
Cell culture	MS medium with 3.5 mg/L NAA and 3% sucrose	Cultures were supplemented with 0.01 mM precursors, such as geranyl pyrophosphate (GPP), geranylgeranyl pyrophosphate (GGPP), isopentenyl pyrophosphate (IPP), dimethylallyl pyrophosphate (DMAPP), farnesyl pyrophosphate (FPP), acetyl- CoA, 3-hydroxy-3-methylglutaryl-CoA (HMG-CoA), glyceraldehyde-3 phosphate (GA-3P), mevalonate (MVA), and sodium pyruvate (SP) after 2 weeks of initial cultures and incubated for a further 5 days	Feeding cultures with IPP as a precursor was the most effective	IPP treatment enhanced bilobalide (BB), ginkgolide A (GA), and ginkgolide B (GB) by 10-, 2.3-, and 6.2-fold in the cells.	[51]
				The IPP treatment also stimulated the excretion of GA and GB by 5.7- and 7.2-fold.	

## 4. Metabolic Engineering for Biosynthesis of Terpene Trilactones 

In general, the overexpression of genes that control the pace of the MEP and MVA pathways regulates the molecular regulation of terpenoid production. These enzymes include 1-deoxy-D-xylulose-5-phosphate reductoisomerase (DXR), 1-deoxy-D-xylulose-5-phosphate synthase (DXS), and (E)-4-hydroxy-3-methylbut-2-enyldiphosphate reductase (HDR), which are involved in the MEP pathway (Figure 3). It is commonly accepted that 3-hydroxy-3-methyl-glutaryl-CoA reductase (HMGR) is the rate-limiting step in the MVA pathway [52]. Other strategies focus on upregulating the prenyltransferases geranyl diphosphate synthase (GPS), farnesyl diphosphate synthase (FPS), or geranylgeranyl diphosphate synthase (GGPPS) to enrich the direct isoprenoid precursor pool. According to Ikram et al. [52], additional strategies include the manipulation of transcription factors or promoters that regulate important genes involved in biosynthesis.

Mevalonate diphosphate decarboxylase (MVD; EC4.1.1.33), which catalyzes the conversion of mevalonate diphosphate to isopentenyl diphosphate, was first identified in *Ginkgo biloba* by Pang et al. [53]. Mevalonate diphosphate was converted to isopentenyl diphosphate (IPP) in *Saccharomyces cerevisiae* strain MN19-34, which was cloned to express this gene.

Leonard et al. [54] used a combination of protein engineering and metabolic engineering to produce levopimaradiene (LP), a precursor of TTLs, in *Escherichia coli*. The mutational enrichment of the GGPPS (S239C/G295D) and levopimaradiene synthase (M5931/Y700F) genes and their expression in *E. coli* augmented the endogenous precursor pools of the GGPPS substrate. Furthermore, this approach made it possible for the target product to have an ideal metabolic flux. Consequently, *E. coli* was modified to produce 700 mg/L of levopimaradiene.

Using *Saccharomyces cervisiae*, Liu et al. [55] produced levopimaradiene (LP), levopimaric acid (LA), and diterpenoid resins. They generated several strains, including t79LPSMM, which was able to produce 23 times more than the strain that contained LPS because of its mutant levopimaradiene synthase. Subsequently, a strain was created by cloning CYP720B1, the enzyme that produces LA from LP, and cytochrome P450 reductases (CPRs) from *Texus cuspidata* (TcCPR) in yeast to assess their LA production capacity. According to Liu et al., [55] the overexpression of the TcPR and CYP720B1 genes at multiple copy locations in the *S. cerevisiae* genome increased LA synthesis in a shake flask culture by 1.9 times to 45.24 mg/L. However, they found that batch-fed fermentation led to further increased production (400.31 mg/L) in 5 L bioreactor cultures.

In a recent study, Forman et al. [27] discovered five multifunctional cytochrome P450s with unusual catalytic activities that produce one of the lactone rings and a tert-butyl group, which are characteristic of all ginkgo trilactone terpenoids.

Zheng et al. [56] identified *Ginkgo biloba* ethylene response factor 4 (GbERF4), which is involved in TTL biosynthesis, in another investigation. The terpenoid content of *Nicotiana tabacum* is significantly increased by overexpressing GbERF4, and key genes involved in terpenoid biosyntheses, such as geranylgeranyl diphosphate synthase [GGPPS], 1-deoxy-D-xylulose-5-phosphate reductoisomerase [DXR], 1-deoxy-D-xylulose-5-phosphate synthase [DXS], acetyl-CoA C-acetyltransferase [AACT], and 3-hydroxy-3-methylglutaryl-CoA reductase [HMGR], are also upregulated, thereby suggesting that GbERF4 functions in regulating the synthesis of terpenoids. MicroRNA (miRNA) sequencing expression pattern analysis revealed that gb-mIR160 inhibited the production of these compounds. Transient expression of GbERF4 enhanced the TTL content in ginkgo, as demonstrated by Zheng et al. [56]. Transcriptome analysis revealed the upregulation of the transcription factors HMGS, DXS, and CYPs. Furthermore, they demonstrated that GbERF4 could bind to the yeast promoters HMGS1, AACT1, DXS1, LPS2, and GGPPS, thus elucidating their function in stimulating their expression. These studies have unequivocally shown that transgenic plants and heterologous systems can produce TTLs more effectively.

Cloning the genes 1-deoxyxylulose-5-phosphate synthase (dxs), isopentenyl-diphosphate delta-isomerase (idi), 4-diphosphocytidyl-2C-methyl-d-erythritol synthase (ispD), 2-C-methyl-d-erythritol 2,4-cyclodiphosphate synthase (ispF), geranylgeranyl diphosphate synthase (ggpps), and levopimaradiene synthase (lps) in *E. coli* is highly advantageous [54] for the production of levopimaradiene, one of the precursors of ginkgolide synthesis. This will make it possible to use simple carbon sources, such as glucose and glycerol, for producing levopimaradiene in *E. coli*. Levopimaradiene may be used to initiate the synthesis of TTLs in *Ginkgo biloba* cell cultures.

An analogous approach involves the cloning of *Ginkgo biloba* cytochrome P450 genes [27], specifically GbCYP7005C1, GbCYP7005C3, GbCYP867E38, GbCYP867K1, and GbCYP720B31, into model plants like *Nicotiana benathamina*. Further ginkgolide synthesis can be anticipated when these transgenic tobacco cell lines are grown in vitro and fed levopimaradiene.

## 5. Conclusions

Terpene trilactones, such as ginkgolides and bilobalides, possess several advantageous biological activities; hence, they are in great demand in the pharmaceutical and cosmetic industries for use as therapeutic agents. The production of TTLs by *Ginkgo biloba* plants in vivo is feasible; however, ginkgo plants are slow-growing and require approximately 30 years to reach maturity. In addition, large variations in their TTL content have been reported with age, sex, and in different parts of the plant. Additionally, variations in the TTL content in plants grown in different geographical regions and seasonalities are another reason for the in vivo production. Therefore, in vitro cell cultures have been adopted for the production of TTLs, and various strategies such as optimization of the culture medium, growth regulators, immobilization, elicitation, and precursor feeding methods have been developed. Efforts have also been made to produce TTLs in heterologous systems using synthetic biology techniques. However, further research is required to understand the complete biosynthetic pathways of TTLs. Thus, the heterologous production of TTLs is only possible using metabolic engineering and synthetic biology.

## Figures and Tables

**Figure 1 plants-13-02575-f001:**
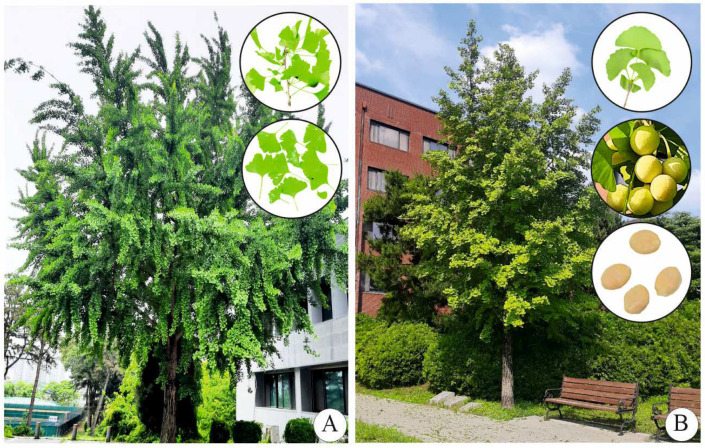
*Ginkgo biloba* trees growing on the campus of Chungbuk National University campus, Cheongju, Korea: male tree (**A**) (twig and leaves shown in circled images); female tree (**B**) (leaves, matured ovules, and seeds shown in circled images).

**Figure 2 plants-13-02575-f002:**
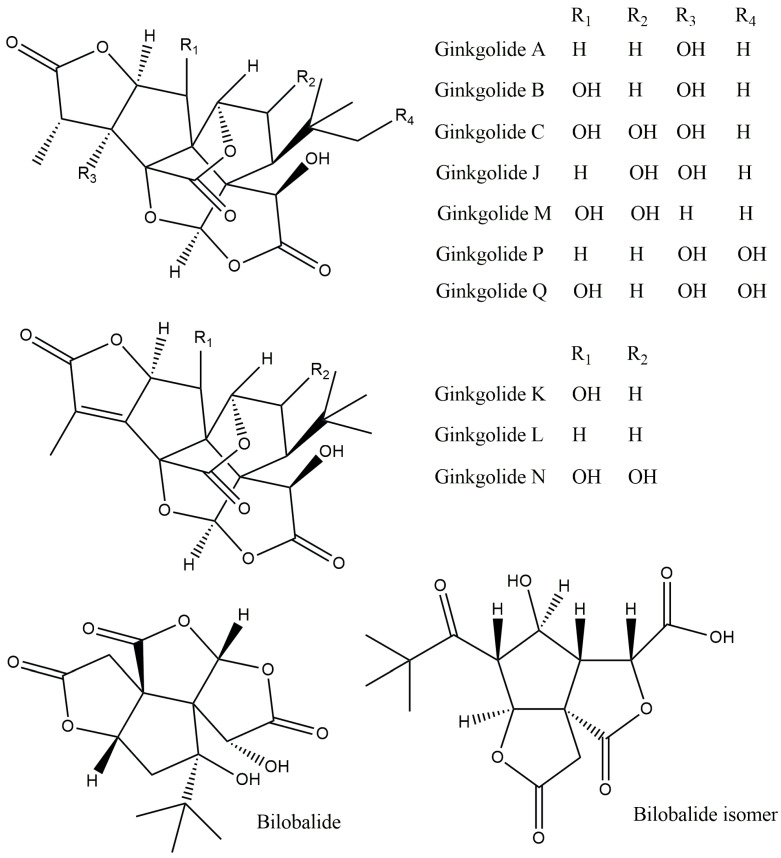
Structures of terpene trilactones (ginkgolides and bilobalide).

**Figure 3 plants-13-02575-f003:**
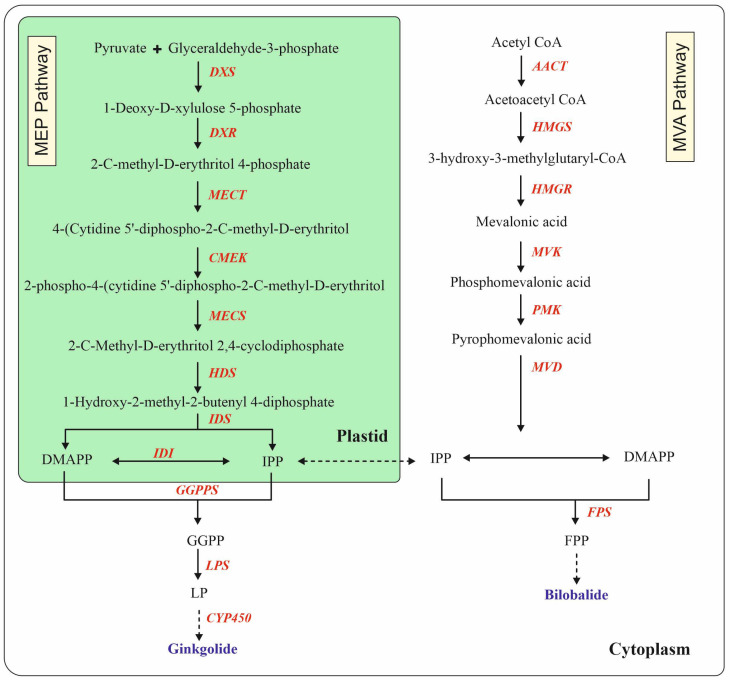
Terpene trilactone biosynthetic pathway. DXS: 1-deoxy-D-xylulose-5-phosphate synthase, DXR: 1-deoxy-Dxylulose-5-phosphate reductoisomerase, MECT: 2-C-methyl-Derythritol 4-phosphate cytidyltransferase, CMEK: 4-(cytidine 50-diphospho)-2-C-methyl-D-erythritol kinase, MECS: 2-Cmethyl-D-erythritol 2,4-cyclodiphosphate synthase, HDS: 1-hydroxy-2-methyl-2-(E)-butenyl 4-diphosphate synthase, HDR: 1-hydroxy-2-methyl-2-(E)-butenyl 4-diphosphate reductase, IDI: isopentenyl-diphosphate delta-isomerase, AACT: acetoactyl-CoA thiolase, HMGS: HMG-CoA synthase, HMGR: HMG-CoA reductase, MVK: mevalonate kinase, PMK: phosphomevalonate kinase, MVD: mevalonate 5-phosphate decarboxylase, LPS: levopimaradiene synthase, CYP 450: cytochrome-P450-dependent monooxygenase. The biosynthetic pathway was created by CoralDRAW^®^ 2021 (version: 23.1.0.389).

## Data Availability

Not applicable.

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
