# Peer review of "Production of Terpene Trilactones from Cell and Organ Cultures of Ginkgo biloba"

_plants, 2024, doi:10.3390/plants13182575_

Round 1

Reviewer 1 Report

Comments and Suggestions for Authors

First of all, it is a pleasure to greet you again. Manuscript  plants-3173283  corresponds to a review article that compiles and describes aspects related to trilactones obtained from Ginkgo biloba extract. In this context, the manuscript does not present a methodology to be evaluated, nor results obtained from it. However, the work is interesting because it compiles useful information on the in vitro culture used in the production of ginkgo terpenoids. Furthermore, the cloning of terpenoid biosynthetic genes to improve terpenoid production is compiled and discussed.

The manuscript is adequately organized into different sections that address some general and other more specific aspects. Among the general aspects, those associated with the biosynthesis of terpene trilactones are described. Perhaps this section might be too general and could be removed or summarized in the introduction.

The rest of the sections are more specific. They compile very useful information regarding the effect of different conditions on the production of terpene trilactones in tissue cultures: influence of nutrients, growth regulators, nitrogen and phosphorus, sucrose, light, temperature, etc. Furthermore, it compiles very useful information about the application of immobilization strategies and the use of precursors to increase the production of terpene lactones. I consider these two sections to represent the best contribution of the work.

Finally, some studies on silencing or overexpression of genes associated with terpene biosynthesis and their influence on the production of terpene lactones are compiled. This section is perhaps too descriptive. I suggest further discussion of this information, highlighting the most promising genes for manipulation in the pursuit of overproduction of terpene lactones.

Currently, I am not studying terpene trilactones from Ginkgo biloba; however, I consider that this manuscript can be  useful for all researchers working or starting studies in this field of research..

The bibliography cited is consistent with the information presented. Below, I detail some observations.

Introduction

Line 31: Write Ginkgo biloba in italic.

Line 31-38: add reference.

Production of terpene trilactones in plant tissue cultures

Lines 123 and 128: Write Ginkgo biloba in italic.

Impact of nitrogen and phosphate levels

Lines 201 and 203: KH2PO4 place the atom numbers as subscripts.

Metabolic engineering for terpene trilactones biosynthesis

Lines 329, 330, 333, 337, 339, 344, 352, 353: Write the scientific names in italic.

Conclusion

Line 370: Write Ginkgo biloba in italic

Author Response

Response to reviewer comments

Manuscript ID: plants-3173283
Type of manuscript: Review
Title: Production of terpene trilactones from cell and organ cultures of Ginkgo biloba

The authors are thankful to anonymous reviewers for their valuable comments on the manuscript and we have revised the manuscript in the light of the reviewer’s comments and incorporated all the corrections suggested by them. Corrections have been included in the revised manuscript in track change format. Following are the specific changes carried out in the revised manuscript.

Reviewer #1

Query #1. The manuscript is adequately organized into section…….

Answer: As per the suggestion of the reviewer the section ‘Biosynthesis of terpene trilactones’ is summarized in the introduction section.

Query #2. Finally, some studies….. I suggest further discussion of this information.

Answer: Discussion on most promising genes for manipulation in the pursuit of overproduction of terpene trilactones has been strengthened as per the suggestion of reviewer.

Query #3. Line 31. Write Ginkgo biloba in italic.

Answer: Correction is incorporated.

Query #4. Add reference.

Answer: Reference is added on the morphological description of ginkgo plants.

Query #5. Line 123 and 129. Write Ginkgo biloba in italic.

Answer: Corrections are incorporated.

Query #6. Lines 201 and 203. KH2PO4 place the atom number as subscripts.

Answer: Corrections are incorporated.

Query #7. Lines 329, 333, 337, 339, 344, 352, 353: Write the scientific names in italic.

Answer: Corrections are incorporated.

Query #8. Conclusion. Line 370: Write Ginkgo biloba in italic.

Answer: Correction is incorporated.

Reviewer 2 Report

Comments and Suggestions for Authors

Line 40: What is the sex of the plant in the picture? Wouldn't it be better to show plants of different sexes than the twig and leaves?

Line 79: How were the structural formulas of the compounds generated? I suggest making this figure larger (e.g. three terpene trilactones on one line and blowing the whole thing up to full page size). I also suggest arranging them in the order shown in Table 1.

Line 80: Please insert some separator in Table 1 (e.g. larger spaces between rows). The table will be more readable because the rows now merge.

Line 116: Please add what literature the engraving was based on and how it was created (e.g. in what program).

Line 242 i Line 312: Table 2 and Table 3, although very interesting and necessary, are not very legible, some rows are centered, rows are not separated, so it is difficult to see where a given example ends. Please re-edit the tables so that they are more legible and not so long.

Author Response

Response to reviewer comments

Manuscript ID: plants-3173283
Type of manuscript: Review
Title: Production of terpene trilactones from cell and organ cultures of Ginkgo biloba

The authors are thankful to anonymous reviewers for their valuable comments on the manuscript and we have revised the manuscript in the light of the reviewer’s comments and incorporated all the corrections suggested by them. Corrections have been included in the revised manuscript in track change format. Following are the specific changes carried out in the revised manuscript.

Reviewer #2

Query #1. Line 40: What is the sex of the plant in the picture? Wouldn’t it be better show plants with different sexes than the twig and leaves?

Answer: As per the suggestion of the reviewer both male and female plants with parts of the plants have been presented in the Figure 1.

Query #2. Line 79. How were the structural formulas of the compounds generate? I suggest making this Figure larger (e.g. three terpene trilactones on one line and blowing the whole thing up to full page size). I also suggest arranging them in order shown in Table 1.

Answer: The chemical structures of the compounds are generated by using ChemDraw®, Revvity Signals Software, Waltham, Massachusetts, United States.

Structures of the compounds have been rearranged in Figure 2 in the line/order presented in Table 1 as suggested by the reviewer. 

Query #3. Line 80. Please insert some separator in Table 1 (e.g. larger spaces between rows). The table will be more readable because the rows now merge.

Answer: The Table 1 is represented as suggested.

Query #4. Line 116: Please add what literature the engraving was based on and how it was created (e.g. in what program).

Answer: The biosynthetic pathway has been drawn by using CoralDRAW® 2021 (version: 23.1.0.389).

Query #5. Line 242, Line 312: Table 2 and Table 3, although very interesting and necessary, are not very legible, some rows are centered, rows are not separated, so it is difficult to see where a give example ends. Please re-edit the tables so that they are more legible and not so long.

Answer: The Table 2 and Table 3 are represented as per suggestion.
